# Cytological and Transcriptomic Analysis Provide Insights into the Formation of Variegated Leaves in *Ilex × altaclerensis* ‘Belgica Aurea’

**DOI:** 10.3390/plants10030552

**Published:** 2021-03-15

**Authors:** Qiang Zhang, Jing Huang, Peng Zhou, Mingzhuo Hao, Min Zhang

**Affiliations:** 1Co-Innovation Center for Sustainable Forestry in Southern China, College of Biology and the Environment, Nanjing Forestry University, 159 Longpan Road, Nanjing 210037, China; zhangqiang@njfu.edu.cn (Q.Z.); hmz@njfu.edu.cn (M.H.); 2Jiangsu Academy of Forestry, 109 Danyang Road, Dongshanqiao, Nanjing 211153, China; xiaojingzi1229@163.com (J.H.); zhoupengjsdt@sina.com (P.Z.)

**Keywords:** *Ilex × altaclerensis* ‘Belgica Aurea’, leaf variegation, chlorophyll, ultrastructure, transcriptome

## Abstract

*Ilex × altaclerensis* ‘Belgica Aurea’ is an attractive ornamental plant bearing yellow-green variegated leaves. However, the mechanisms underlying the formation of leaf variegation in this species are still unclear. Here, the juvenile yellow leaves and mature variegated leaves of *I. altaclerensis* ‘Belgica Aurea’ were compared in terms of leaf structure, pigment content and transcriptomics. The results showed that no obvious differences in histology were noticed between yellow and variegated leaves, however, ruptured thylakoid membranes and altered ultrastructure of chloroplasts were found in yellow leaves (yellow) and yellow sectors of the variegated leaves (variegation). Moreover, the yellow leaves and the yellow sectors of variegated leaves had significantly lower chlorophyll compared to green sectors of the variegated leaves (green). In addition, transcriptomic sequencing identified 1675 differentially expressed genes (DEGs) among the three pairwise comparisons (yellow vs. green, variegation vs. green, yellow vs. variegation). Expression of magnesium-protoporphyrin IX monomethyl ester (MgPME) [oxidative] cyclase, monogalactosyldiacylglycerol (MGDG) synthase and digalactosyldiacylglycerol (DGDG) synthase were decreased in the yellow leaves. Altogether, chlorophyll deficiency might be the main factors driving the formation of leaf variegation in *I.*
*altaclerensis* ‘Belgica Aurea’.

## 1. Introduction

Variegated leaves are the leaves with regular or irregular non-green spots and patches on the leaf surface [1]. This special attractive trait increases the economic value of ornamental plants, and the variegated leaf plants have been widely used in urban landscaping. Foliar variegation has been considered as a kind of defensive plant coloration [2], which may defend leaves from herbivore damage by camouflage, aposematism [3]. However, the defensive mechanisms of foliar variegation were not fully understood. It was also proposed that apart from protective role, leaf variegation had some potential physiological advantages. The work of Shelef et al. showed that white variegation of *Silybum marianum* could elevate leaf temperature during cold winter days [4]. Further study revealed that low temperature protective function of leaf variegation was partly associated with increased ROS-scavenging enzymatic activity [5].

A recent study based on the investigation of 1710 species with variegated leaves belonging to 78 families divided the variegated leaves into five types: chlorophyll type, air space type, epidermis type, pigment type and appendages type [1]. In addition, a four-type classification concept (total chlorophyll increased type, total chlorophyll deficient type, chlorophyll a deficient type and chlorophyll b deficient type) was also recommended to classify leaf color mutants [6].

Leaf variegation could be induced by gene mutations related to chloroplast biogenesis or photosynthetic pigments’ biosynthesis [7]. The molecular mechanisms of leaf variegation have been deeply investigated in *Arabidopsis* mutants. It was reported that leaf-variegated mutation *var1* and *var2* of *Arabidopsis* were caused by nuclear recessive mutation in *FtsH* genes, resulting in disrupted plastids and formation of green/white sectors [8]. Further study revealed that balance between cytosolic and chloroplast translation could control the extent of *var2* variegation [9]. Another *Arabidopsis* variegation mutant *immutans* (*im*) was formed by mutation of *im*, a gene encoding the terminal oxidase of chlororespiration [7]. Mutation in the DNAJ-like protein encoding gene *SNOWY COTYLEDON 2* (*SCO2*)/*CYO1* was reported to cause chlorotic cotyledons but green true leaves [10]. Whereas, a recent study noticed that mutations in *SCO2* could affect development of true leaves in *Arabidopsis* under short-day conditions, and induce leaf variegation in *Lotus japonicas* [11]. In plant species other than *Arabidopsis*, it was found that mutation in chlorophyll a apoprotein A1 gene could lead to variegation in *Helianthus annuus* L. [12]. In *Cymbidium sinense*, the ethylene response factors (ERFs) gene *CsERF2* played crucial roles in leaf variegation [13]. Moreover, data showed that disruption of carotene biosynthesis in *Brassica napus* also led to abnormal plastids and variegated leaves [14].

*Ilex* L. is the sole genus of the family Aquifoliaceae, comprises approximately 600 species of dioecious trees and shrubs [15,16]. *Ilex × altaclerensis* ‘Belgica Aurea’ is a cultivar belonging to Belgica group, one of the six groups under *Ilex × altaclerensis* [17]. *I. altaclerensis* ‘Belgica Aurea’ is a small, evergreen tree or shrub which bears dark green leaves edged with creamy-yellow. In autumn and winter, *I. altaclerensis* ‘Belgica Aurea’ has sparse red berries on the branches. Due to these attracting characteristics, *I. altaclerensis* ‘Belgica Aurea’ has been used as ornamental plants in gardens and parks.

To elucidate the regulatory mechanisms for leaf variegation, leaf structure, pigment content and transcriptomics were compared between juvenile yellow leaves and variegated leaves of *I. altaclerensis* ‘Belgica Aurea’ in the present study. The assembled unigenes were classified by GO and KEGG analysis, DEGs related to chlorophyll metabolism, and chloroplast integrity and function were identified. Moreover, quantitative real-time PCR (qRT-PCR) were performed to validate RNA-Seq results. Our findings may provide a useful reference for characterizing leaf variegation in the woody plants.

## 2. Results

### 2.1. Leaf Anatomical Characteristics and Ultrastructure

The juvenile yellow leaf (Figure 1A) and mature variegated leaf (Figure 1C) were transversely sectioned. The leaf blade anatomy revealed that the epidermis in both surfaces of yellow and variegated leaves are two-layered (Figure 1B,D). The palisade tissue and spongy tissue were obviously differentiated in the yellow leaves (y), whereas, few chloroplasts were found in mesophyll tissue (Figure 1B). The leaves of *I. altaclerensis* ‘Belgica Aurea’ exhibited morphological characteristics of heliophyte with approximately three layers of palisade parenchyma cells and nine layers of loosely arranged spongy parenchyma cells. The intercellular airspaces were distributed in the spongy tissue. No obvious differences in leaf anatomy were observed between the variegated leaves and the yellow leaves. The leaf structure of yellow parts (v) and green parts (g) of the variegated leaves were similar, but a great number of chloroplasts were present in the chlorenchyma of green parts (Figure 1D).

Transmission electron microscope (TEM) observations showed that the chloroplasts were ellipsoidal shape. The chloroplasts in the mesophyll cell from green parts (g) of the variegated leaves were well developed with complete grana stacks and well-arranged stroma lamellae (Figure 2A). In contrast, only few thylakoid lamellae were found in the chloroplasts of yellow leaves (Figure 2B) and yellow parts of the variegated leaves (Figure 2C), and no complete grana stacks were formed.

### 2.2. Pigment Content in Yellow and Variegated Leaves

Pigment contents in yellow and variegated leaves were quantified spectrophotometrically (Table 1). The content of chlorophyll a, chlorophyll b, total chlorophyll (chlorophyll a+b) and carotenoid in the green part of variegated leaves was significantly higher than those of yellow leaves and the yellow part of variegated leaves. Compared with the green part of variegated leaves, the content of total chlorophyll in yellow leaves and the yellow part of variegated leaves were dramatically decreased by 87.35% and 91.57%, respectively. The chlorophyll a/b ratio and total chlorophyll/carotenoid were also significantly high in the green parts. However, no significant differences in the content of chlorophyll a, total chlorophyll and carotenoid between yellow leaves and the yellow part of variegated leaves was found. In addition, the level of anthocyanin between the green and yellow parts was significantly different, whereas the difference between yellow parts of the variegated leaves and yellow leaves was not significant.

### 2.3. Localization of Pigments in the Leaves

In Figure 3, the in situ localization of chlorophyll, carotenoids and anthocyanins in leaf sections was observed by laser confocal microscopy. Only very dim red chlorophyll fluorescence was observed in the juvenile yellow leaves, while strong intensity of chlorophyll fluorescence was present in the green parts of the variegated leaves. Especially in the palisade tissue, accumulation of chloroplasts was found. In the yellow parts of the variegated leaves, the intensity of chlorophyll fluorescence was also very weak, just as that in the juvenile yellow leaves. It was noticed that the carotenoids’ fluorescence in the chlorenchyma was extremely faint both in the yellow leaves and the variegated leaves. The autofluorescence of anthocyanins in yellow leaves was weak, and the distribution pattern and intensity of anthocyanins’ fluorescence in the variegated leaves showed a similar pattern as that in the yellow leaves.

### 2.4. RNA-Seq Analysis

A total of nine samples (three biological replicates for each group) were sequenced generating 78,637 unigenes with an average length of 1536 bp after filtering the low quality reads. The size distribution of these unigenes was displayed in Appendix A. The median contig (N50) length and GC% of sequenced data was 2296 bp and 41.37%, respectively. Further, 83.36–87.55% of these reads were mapped to the reference genome. Summary of the sequencing and assembly was shown in Appendix A. The sequencing raw data have been deposited in Sequence Read Archive database of the National Center for Biotechnology Information (accession number: PRJNA680216).

### 2.5. Gene Annotation and Function Classification

A total of 78,637 assembled sequences were BLAST-searched against Nr, Nt, Swiss-Prot, KEGG, KOG, Pfam and GO databases, resulting in 62,275 annotated unigenes, which accounted for 79.19% of the assembled sequences (Table 2). Among the annotated unigenes, 59,752 unigenes were annotated to Nr and 52,105 unigenes could be annotated to Nt, accounting for 75.98% and 66.26% of all unigenes, respectively, and 45,066 (57.31%) unigenes were assigned in Swiss-Prot with 46,101 (58.63%) unigenes annotated by Pfam.

Using the Blast2GO program, 34,793 (44.25%) unigenes were categorized as ‘biological process’, ‘cellular component’ or ‘molecular function’. These three major categories were subdivided into 39 GO classes including 15 ‘biological process’, 11 ‘cellular component’ and 13 ‘molecular function’ (Figure 4, Appendix A). In the ‘biological process’, ‘cellular process’ (10,639 unigenes, 42.42%), ‘biological regulation’ (4339 unigenes, 17.30%) and ‘localization’ (2554 unigenes, 10.18%) were the three prominent subclasses. For ‘cellular components’, ‘cell’ (11,144 unigenes, 42.20%), ‘membrane part’ (9556 unigenes, 36.18%) and ‘organelle part’ (4365 unigenes, 16.53%) were dominant. Furthermore, ‘binding’ (17,126 unigenes, 44.85%) and ‘catalytic activity’ (16,384 unigenes, 42.91%) were prominently represented in ‘molecular function’.

The KEGG analysis showed that 47,967 unigenes participated in five metabolic pathways, which were ‘cellular processes’, ‘environmental information processing’, ‘genetic information processing’, ‘metabolism’ and ‘organismal systems’ (Figure 5, Appendix A). ‘Global and overview maps’ was the largest class (10,607 unigenes, 24.03%) followed by ‘carbohydrate metabolism’ (9.66%, 4266 unigenes).

To classify orthologous gene products, 47,369 unigenes (60.24% of all unigenes) were divided into 25 KOG classifications (Figure 6, Appendix A). Among them, ‘general function prediction only’ was the largest group (10,037 unigenes, 21.60%) and ‘signal transduction mechanisms’ ranked the second one (4904 unigenes, 10.55%).

### 2.6. Gene Expression Analysis

Changes in gene expression were determined by comparing yellow leaves (yellow) vs. the green part of the variegated leaves (green), the yellow part (variegation) vs. green part of the variegated leaves, and yellow vs. variegation. A total of 18,415 genes were differentially expressed between yellow and green, among which 10,535 genes were upregulated and 7880 genes were downregulated (Figure 7A). Between variegation and green, the expression level of 16,883 genes significantly changed, including 8966 upregulated genes and 7917 downregulated genes. In contrast, only 6660 genes were identified differentially expressed between yellow and variegation. Of these DEGs, 3673 were upregulated and 2987 were downregulated. Based on the analysis of three pairwise comparisons (yellow vs. green, variegation vs. green, yellow vs. variegation), 1675 DEGs were found to be shared by all three DEG sets in Venn diagram (Figure 7B, Appendix A).

In comparison to the shared DEGs, 4240, 3120 and 877 DEGs were differentially expressed in only yellow vs. green, variegation vs. green, and yellow vs. variegation, respectively. Moreover, hierarchical clustering of the total 1675 DEGs based on the similarity of gene expression patterns revealed that yellow vs. green and variegation vs. green clustered together, while yellow vs. variegation clustered distantly from these two groups (Figure 7C).

### 2.7. GO Enrichment Analysis of DEGs

To further investigate the functional information of DEGs, the DEGs expressed in the overlap of the three pairwise comparisons were subjected to GO and KEGG enrichment analysis. Based on GO classification, the DEGs were classified into three main categories: ‘molecular function’, ‘cellular component’ and ‘biological process’, among which 1122 DEGs were assigned to the ‘molecular function’ category, 1533 DEGs were assigned to the ‘cellular component’ and 1196 DEGs were assigned to the ‘biological process’. These DEGs were further classified into 45 functional subcategories (Appendix A).

In ‘biological process’, ‘cell process’ (320 DEGs) and ‘metabolic process’ (354 DEGs) were the most abundantly represented subcategories, whereas, ‘cell proliferation’, ‘carbon utilization’ and ‘detoxification’ were the smallest ones—only one DEGs for each of these three subcategories. ‘Membrane’ (323 DEGs) and ‘membrane part’ (306 DEGs) were dominant in the ‘cellular component’. For ‘molecular function’, the DEGs were mainly involved in ‘catalytic activity’ (507 DEGs) and ‘binding’ (448 DEGs).

The GO functional enrichment analysis revealed that the most enriched terms of ‘cellular component’ were ‘integral component of membrane’ (GO: 0016021) which involved 295 DEGs (Figure 8, Appendix A). ‘ATP binding’ (GO: 0005524), ‘metal ion binding’ (GO: 0046872) and ‘RNA binding’ (GO: 0003723) were the three most significant GO enrichment terms involved in ‘molecular function’. In addition, one GO term was identified to be related to photosynthesis and light harvesting process, i.e., ‘chlorophyll binding’ (GO: 0016168). In ‘biological process’, the DEGs were primarily enriched for the terms ‘transcription, DNA-templated’ (GO: 0006351), ‘carbohydrate metabolic process’ (GO: 0005975) and ‘metabolic process’ (GO: 0008152). Among these GO terms, ‘chlorophyll biosynthetic process’ (GO: 0015995) and ‘chlorophyll catabolic process’ (GO: 0015996) were closely associated with chlorophyll metabolism.

### 2.8. KEGG Enrichment Analysis of DEGs

To identify the specific biochemical pathways involved in the formation of leaf variegation, the DEGs were subjected to KEGG pathway enrichment analysis (Figure 9, Appendix A). The DEGs were mapped to 123 pathways in the KEGG database. Among these 123 KEGG pathways, most of the DEGs (66.29% of all the DEGs) were related to ‘metabolism’. The most significantly enriched pathway was phenylpropanoid biosynthesis (Ko00940, 64 DEGs, 4.85%) and plant-pathogen interaction (Ko04626, 64 DEGs, 4.85%), followed by starch and sucrose metabolism (ko00500, 57 DEGs, 4.32%), MAPK signaling pathway-plant (ko04016, 39 DEGs, 2.95%) and carbon metabolism (ko01200, 36 DEGs, 2.73%). Moreover, 20 DEGs were found to be involved in carbon fixation in photosynthetic organisms (ko00710), and 6 DEGs were involved in photosynthesis (ko00195). One DEG was identified as photosynthesis-antenna proteins (ko00196) and 17 DEGs were associated with oxidative phosphorylation (ko0019). These results suggested that energy metabolism was different between the yellow and variegated leaves.

### 2.9. Analysis of the Genes Involved in Chlorophyll Metabolic Pathway and Chloroplast Integrity and Function

Chlorophyll catabolism plays a critical role in determining leaf coloration. KEGG enrichment analysis revealed that a total of 6 DEGs were closely related to porphyrin and chlorophyll metabolism (ko00860) (Table 3). The encoding gene of magnesium-protoporphyrin IX monomethyl ester [oxidative] cyclase (CL5332.Contig4_All) was found to be downregulated in yellow leaves and yellow parts of the variegated leaves. Two unigenes (CL374.Contig9_All and CL8272.Contig2_All) encoding chlorophyll(ide) b reductase were also downregulated in yellow leaves and yellow parts of the variegated leaves, while one unigene (CL8272.Contig3_All) encoding this enzyme was upregulated. Chlorophyll(ide) b reductase is one of the two key enzymes involved in the conversion of chlorophyll b to chlorophyll a, which is a crucial step in chlorophyll catabolism [18]. The transcriptome data also showed that the expression of monogalactosyldiacylglycerol synthase (Unigene31820_All) and digalactosyldiacylglycerol synthase (CL3869.Contig6_All) were downregulated in yellow leaves and yellow parts of the variegated leaves.

### 2.10. Validation of RNA Sequencing Data by qRT-PCR

To validate the RNA-Seq results, 7 DEGs were selected to conduct qRT-PCR analysis using the same RNA samples for RNA-Seq analysis. These DEGs encoded the following enzymes: magnesium-protoporphyrin IX monomethyl ester [oxidative] cyclase (CL5332.Contig4_All), chlorophyll(ide) b reductase (CL374.Contig9_All), monogalactosyldiacylglycerol synthase (Unigene31820_All), digalactosyldiacylglycerol synthase (CL3869.Contig6_All), photosystem II oxygen-evolving enhancer protein 1 (CL782.Contig2_All), 9-cis-epoxycarotenoid dioxygenase (Unigene6311_All) and abscisic acid 8′-hydroxylase 3-like (CL1494.Contig2_All). The qRT-PCR data showed that the relative expression levels of all 7 DEGs increased in the green parts of the variegated leaves (Figure 10). Such results indicated that the transcriptome analysis was reliable.

## 3. Discussion

Leaf variegation is a phenotype frequently observed in higher plants, which occurs naturally or by mutagenesis [19]. The most striking characteristic of variegated leaf is the formation of white or yellow and green sectors in the same leaf. The green tissue contains normal well developed chloroplasts, while in the white sectors, chloroplasts are impaired [19]. Thus, leaf-variegated mutants have attracted special attention from plant scientists to use them in exploring the mechanisms of chloroplast development [7]. Using model species, especially *Arabidopsis*, great progress has been achieved in the understanding of variegations at the molecular level.

To elucidate the mechanisms of variegation formed in the leaves of *I. altaclerensis* ‘Belgica Aurea’, the anatomy of yellow and variegated leaves was performed. The histological results showed no observed structure differences between yellow and variegated leaves. Moreover, no intercellular air spaces found between the epidermal and palisade cells in yellow leaves and yellow parts of the variegated leaves. Thus, according to the classification of leaf variegation types recommended by Zhang et al. [1], *I. altaclerensis* ‘Belgica Aurea’ is chlorophyll type variegation. However, there was some differences between the variegated leaves of *I. altaclerensis* ‘Belgica Aurea’ and *Goeppertia makoyana*, a species studied by Zhang et al. [1]. In the yellow parts of *I. altaclerensis* ‘Belgica Aurea’ variegated leaves, chloroplasts rarely presented in mesophyll tissue; while chloroplasts in discolored leaf area of *G. makoyana* were identical as those in green area, and the discolored parts of the leaves were light green or yellowish green.

The structural integrity of thylakoid membranes is essential for chloroplast function. Thylakoid membranes provide a platform for photosynthetic protein pigment complexes, and the conversion of energy by photosynthesis occurs there [20]. Previous studies have reported that the fine structure of chloroplasts altered in the yellow sectors of variegated leaves. For example, chloroplasts in the green sectors of *H. annuus* L showed well-organized stromal and granal thylakoids, while plastids in the yellow sectors only contained poorly developed granum-stroma thylakoid membrane system [12]. In our study, TEM revealed a strong contrast in chloroplast ultrastructure between juvenile yellow leaves, yellow sectors and green sectors of the variegated leaves. In yellow leaves and yellow sectors of the variegated leaves, the ultrastructure of chloroplasts was severely altered with only small number of thylakoid lamellae and the arrangement was also disordered. In contrast, thylakoids in the chloroplasts of the green sectors were well-organized, suggesting that the chloroplasts were well developed in green sectors of the mature variegated leaves. Similar results were also found in the variegated leaves of *Phalaenopsis aphrodite* subsp. formosana [21], leaves of *Ginkgo biloba* gold-colored mutant and yellow leaf of *Quercus shumardii* Buckley [22,23].

Deficiency in chlorophyll has been reported in many leaf color mutations, such as *Arabidopsis*, *Ananas comosus* var. bracteatus, *Helianthus annuus* L., etc. [12,22,24,25]. In the present study, the content of chlorophyll a, chlorophyll b, total chlorophyll in the yellow leaves and the yellow sectors of variegated leaves were all significantly lower than that of green sectors. So, the leaf variegation formed in *I. altaclerensis* ‘Belgica Aurea’ was total chlorophyll deficient type. Our results were somewhat different from the research of Li et al. [22]. They suggested that lack of chlorophyll b contributed to golden leaf coloration in *G. biloba*. Another finding in our research was that the content of carotenoids was also low in the yellow leaves and the yellow sectors of variegated leaves. This result was similar to that in the yellow leaf sectors of the *Var1* and *Var33* mutants of *Helianthus annuus* L. [12].

Transcriptomic analysis using RNA-Seq has been widely used to identify genes that are differentially regulated at various developmental stages or in different physiological conditions. RNA-Seq has also been used to elucidate variegation mechanisms. In a previous study, transcriptome analysis revealed that the genes related to photosynthesis and chloroplast functions were repressed in the white sectors of yellow variegated2 (*var2*) mutant in *Arabidopsis* [26]. Furthermore, in the gold-colored mutant leaves of *G. biloba*, the chlorophyll biosynthesis-related Protoporphyrinogen IX oxidase (PPO) showed expressional repression, while chlorophyll degradation-related chlorophyll b reductase (NYC/NOL) had up-regulated expression [22]. The illumina sequencing results revealed a total of 1675 DEGs by comparing yellow leaves with green and yellow sectors of variegated leaves. The biosynthesis of chlorophyll is catalyzed by 15 enzymes, any obstacle in this process will lead to chlorophyll deficient and leaf color mutant [6]. Additionally, 6 DEGs closely related to porphyrin and chlorophyll metabolism were identified based on KEGG pathway assignments. Among these DEGs, the expression levels of magnesium-protoporphyrin IX monomethyl ester (MgPME) [oxidative] cyclase were dramatically reduced in the yellow leaves and the yellow sectors of variegated leaves. MgPME cyclase is an essential enzyme involved in chlorophyll biosynthesis. Loss-of-function mutation in MgPME cyclase resulted in *yellow-green leaf* 8 (*ygl8*) mutant in rice [27]. In *ygl8*, mutant chlorophyll biosynthesis and chloroplast development in young leaves were affected. Moreover, expression of monogalactosyldiacylglycerol (MGDG) synthase and digalactosyldiacylglycerol (DGDG) synthase were suppressed in yellow leaves and yellow parts of the variegated leaves. In plants, the two galactolipids MGDG and DGDG are the major lipids of photosynthetic membranes, constituting approximately 75% of total thylakoid membrane lipids in leaves [28,29]. MGDG and DGDG are major structural components of the thylakoid membrane. Thus, alteration in lipid composition strongly affect the structure and characteristics of the thylakoid membrane [30]. It had been reported that lack of both galactolipids MGDG and DGDG caused severe defects in chloroplast biogenesis [31]. Therefore, abnormal development and function of plastids might play a role in leaf variegation of *I. altaclerensis* ‘Belgica Aurea’. Additionally, the expression level of two unigenes encoding chlorophyll(ide) b reductase were lowered in yellow leaves and yellow parts of the variegated leaves, while one unigene encoding this enzyme was upregulated. These three unigenes encoding chlorophyll(ide) b reductase may represent different transcripts of a single gene or different genes of a gene family. Furthermore, analysis of the 1675 DEGs revealed that the most upregulated gene in green sectors of the variegated leaves was Unigene30740_All, which encoded the DNA-directed RNA polymerase alpha subunit (chloroplast) (Appendix A). This may suggest that the elevated expression of RNA polymerase mRNA in chloroplasts was involved in chloroplast biogenesis. The most downregulated gene in green sectors of the variegated leaves was CL222.Contig6_All encoding NAD(P)H-quinone oxidoreductase subunit 2. The reason of the downregulation of this gene needs further investigation. Altogether, it suggested that the genes involved in chlorophylls’ biosynthesis and chloroplast biogenesis were suppressed in the juvenile yellow leaves. When the leaves became matured, these genes gained their function in some areas of the leaves and thus formed variegation. However, the molecular mechanisms behind need further investigation.

## 4. Materials and Methods

### 4.1. Plant Materials

*I. altaclerensis* ‘Belgica Aurea’ was cultured in the nursery in Jiangsu Academy of Forestry, China. This plant has full yellow juvenile leaves and mature variegated leaves (Figure 1A,C). The yellow patches at the margin of the variegated leaves exhibit marked heterogeneity among different leaves. The plants grown in the nursery received a maximum photosynthetic photon flux density (PPFD) of approximately 1500 μmol m^−2^ s^−1^ and were irrigated regularly with tap water.

### 4.2. Measurements of Contents of Chlorophyll, Carotenoids and Anthocyanins

Approximately 0.2 g (fresh weight) of leaf tissue from juvenile yellow leaves (y), green sector (g) and yellow sector (v) of mature variegated leaves were submerged in 80% acetone, placed in the dark for 24 h at 4 °C. Then, the extracted pigments were measured spectrophotometrically at 663.2, 646.8, and 470 nm, and the contents of chlorophyll and carotenoids were estimated according to Lichtenthaler [32]. To measure the content of anthocyanins, the excised leaves were put into 3 mL methanol-HCl solution (6 M HCl: H_2_O: MeOH = 7:23:70) for 24 h at 4 °C. The extracts were assayed using a spectrophotometer following the method of Hughes et al. [33].

### 4.3. Transmission Electron Microscopic Observation

Samples dissected from the freshly harvested juvenile yellow leaves and mature variegated leaves were cut into small segments (1 mm × 2 mm). The tissues were prefixed in 4% glutaraldehyde in 0.1 M phosphate buffer (pH 7.2) for 24 h at 4 °C. After being rinsed with 0.1 M phosphate buffer (pH 7.2) thrice, tissues were postfixed with 1% OsO_4_ for 4 h at 4 °C, followed by three times rinsing with phosphate buffer (pH 7.2). The samples were then dehydrated in a graded ethanol series (50%, 70%, 80%, 90% and absolute ethanol), 20 min for each step. The dehydrated tissues were embedded in Epon 812 resin and ultrathin sections were cut and stained with 1% uranyl acetate and 1% lead citrate. The sections were observed and photographed using a JEM-1400 transmission electron microscope (JEOL, Tokyo, Japan).

### 4.4. Leaf Histology

For the characterization of leaf architecture, the juvenile yellow leaves and mature variegated leaves were histologically analyzed. Briefly, dissected leaf samples were embedded in OCT compound (Sakura Finetek, CA, USA) and sections (20 μm thick) were obtained in a Leica CM1950 cryostat. The obtained sections were adhered to microscope slides pretreated with poly-L-lysine. The samples were then observed and photographed using a light microscope (Nikon Eclipse 50i, Tokyo, Japan) coupled to an image capture system.

### 4.5. Laser Scanning Confocal Microscopy

Laser confocal microscopy was performed on a Leica TCS SP5 confocal scanning microscope (Leica Microsystems, Heidelberg GmbH, Mannheim, Germany) to investigate distribution of chlorophyll, carotenoids and anthocyanins in the leaves. Leaf samples were dissected and placed in embedding medium (Tissue-tek OCT compound, Sakura Finetek, CA, USA). A Leica CM1950 cryostat was employed to cut the embedded tissue samples and 10 μm thickness transverse sections were made. The sections were transferred to poly-L-lysine-coated slides for observation. For collecting chlorophyll fluorescence, a filter with 633 nm excitation and 650–700 nm emission was used, and for carotenoid autofluorescence, a filter with 488 nm excitation and 500–600 nm emission bands was used [34]. Anthocyanins were excited at 543 nm with a helium-neon laser [35]. DAPI (4′,6′-diamidino-2-phenylindole) was used to counterstain the tissues.

### 4.6. RNA Extraction and cDNA Library Preparation

The leaves were harvested between 9:00–12:00 a.m. and put into liquid N_2_ immediately. Then, the leaf samples were transferred to the lab for RNA extraction. Total RNA was extracted from the leaves of three biological replicates using RNeasy Plant Mini Kit (Qiagen, Hilden, Germany) following the instructions of the manufacturer. The quality of RNA samples was inspected with a Nanodrop 1000 spectrophotometer (Thermo Fisher Scientific, Inc., Waltham, MA, USA), and an Agilent 2100 Bioanalyzer (Agilent Technologies, Inc., Santa Clara, CA, USA) was used to examine RNA integrity and to quantify RNA concentration.

mRNAs of each sample were enriched using oligo (dT)-attached magnetic beads. The oligo(dT)-magnetic beads bound to poly(A)+tails of mRNA and the bound mRNA was pelleted using a magnet. The enriched and purified mRNAs were then broken into short fragments, which were employed as templates to synthesize cDNA. The first strand of cDNA was synthesized using random hexamer primers and Reverse Transcriptase (Qiagen, Hilden, Germany), and the second strand synthesis was performed using DNA Polymerase I (New England BioLabs, Ipswich, MA, USA) and RNase H. The double-stranded cDNA fragments were ligated to the adapter with ‘T’ at 3′ end. The ligation products were then subjected to PCR amplification using High-Fidelity DNA polymerase (Takara Bio, Dalian, China). Finally, the cDNA libraries were sequenced using BGISEQ-500 sequencing platform (Shenzhen, China) and data were provided by the Beijing Genomics Institute (BGI).

### 4.7. Transcriptome Assembly and Functional Annotation

The original sequencing data were transformed into raw reads by base calling. Raw reads were filtered by SOAPnuke (v.1.4.0; https://github.com/BGI-flexlab/SOAPnuke, settings: -I5-q0.5-n0.1, accessed on 20 November 2018) and quality-trimmed with Trimmomatic (v.0.3.6, settings: ILLUMINACLIP: 2:30:10 LEADING: 3 TRAILING:3 SLIDINGWINDOW: 4:15 MINLEN:50) to remove reads containing adapters [36,37], reads with more than 5% unknown bases, and low quality reads (more than 20% bases with small Qphred ≤ 10). The obtained clean reads were assembled using Trinity v.2.0.6 and mapped to reference genomes using Bowtie2 software (v.2.2.5; http://bowtie-bio.sourceforge.net/Bowtie2/index.shtml, accessed on 20 November 2018) [38]. Function of the assembled unigenes was then annotated based on the following seven databases: NCBI non-redundant protein sequences (Nr), NCBI non-redundant nucleotide sequences (Nt), Protein family (Pfam), Clusters of Orthologous Groups of proteins (KOG), Kyoto Encyclopedia of Genes and Genomes (KEGG), Swiss-Prot (a manually annotated and reviewed protein sequence database) and Gene Ontology (GO).

### 4.8. Analysis of Differentially Expressed Genes (DEGs) 

Clean data were mapped back onto the assembled transcriptome, and gene expression levels were estimated by RSEM for each sample [39]. Differential expression analysis between different samples was performed using the DEGseq R package based on the RPKM value (reads per kb per million reads) [40]. Genes with fold change ≥2 and adjusted *p* value ≤ 0.001 were considered as differentially expressed. The DEGs identified were used for further GO and KEGG enrichment analysis. GO enrichment analysis of DEGs was performed using the GOseq R package [41], and the statistical enrichment of DEGs in KEGG pathways was conducted using KOBAS software [42].

### 4.9. Quantitative Real-Time PCR

Validation of the DEGs identified from transcriptome sequencing was carried out by quantitative real-time PCR (qRT-PCR) analysis. Approximately 0.2 μg of total RNA was transcribed into cDNA with a TransScript^®^ First-Strand cDNA Synthesis SuperMix (TransGen Biotech, Beijing, China). qRT-PCR was performed using a LightCycler^®^480 system (Roche, Basel, Switzerland) with SuperReal PreMix Plus (SYBR Green) (TianGen, Beijing, China). Amplification was conducted using the following program: 95 °C for 15 s, 40 cycles of 95 °C for 10 s, 55 °C for 20 s, followed by 72 °C for 20 s. The β-actin gene was used as an internal reference for qRT-PCR detection. Relative mRNA levels were calculated by 2^−∆∆CT^ method. All primers used in the study were designed by Primer3 (v. 0.4.0) and presented in Appendix A [43,44].

### 4.10. Statistical Analysis

The data obtained were analyzed using SigmaStat software (Version 3.5). One-way ANOVA was performed for multiple comparisons, and data were presented as mean ± SEM of three replicates.

## Figures and Tables

**Figure 1 plants-10-00552-f001:**
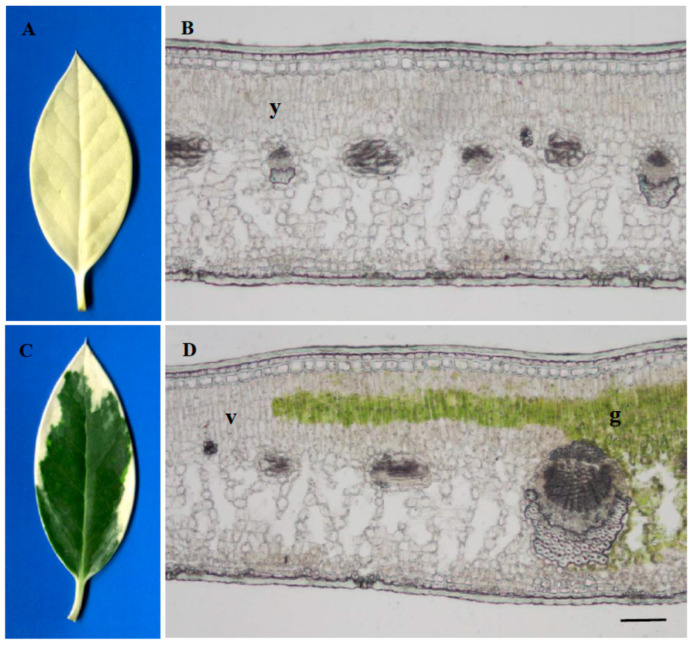
Comparative leaf structures between the full-yellow juvenile leaf (**A**) and mature variegated leaf (**C**) of *I. altaclerensis* ‘Belgica Aurea’. (**B**,**D**) were transverse sections of full-yellow and variegated leaf, respectively. Scale bar = 100 μm. y: full-yellow juvenile leaf; v: yellow sectors of variegated leaf; g: green sectors of variegated leaf.

**Figure 2 plants-10-00552-f002:**
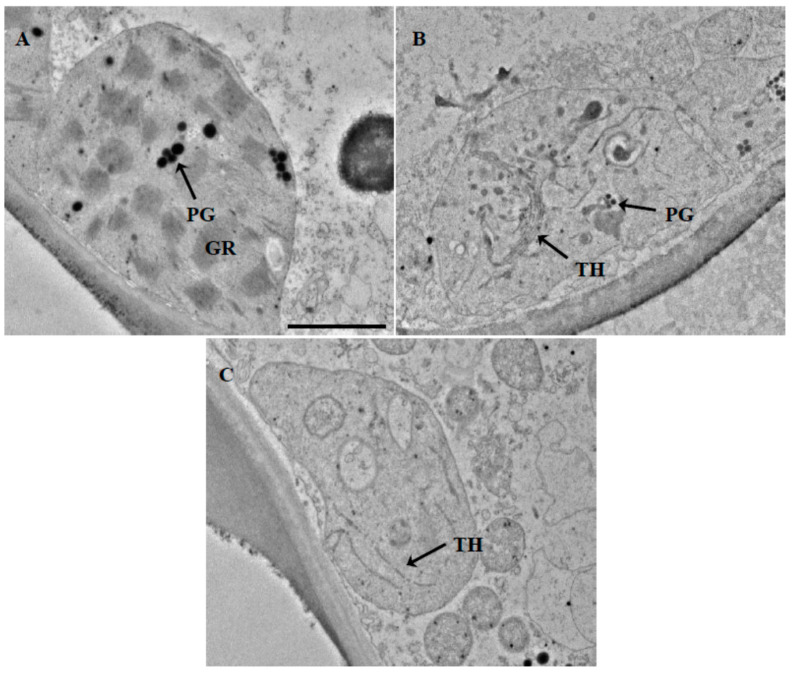
Ultrastructure of mesophyll cells from green (**A**) and yellow (**C**) sectors of variegated leaf, and the full-yellow juvenile leaf (**B**). GR: grana; TH: thylakoid; PG: plastoglobule. Scale bar = 2 μm.

**Figure 3 plants-10-00552-f003:**
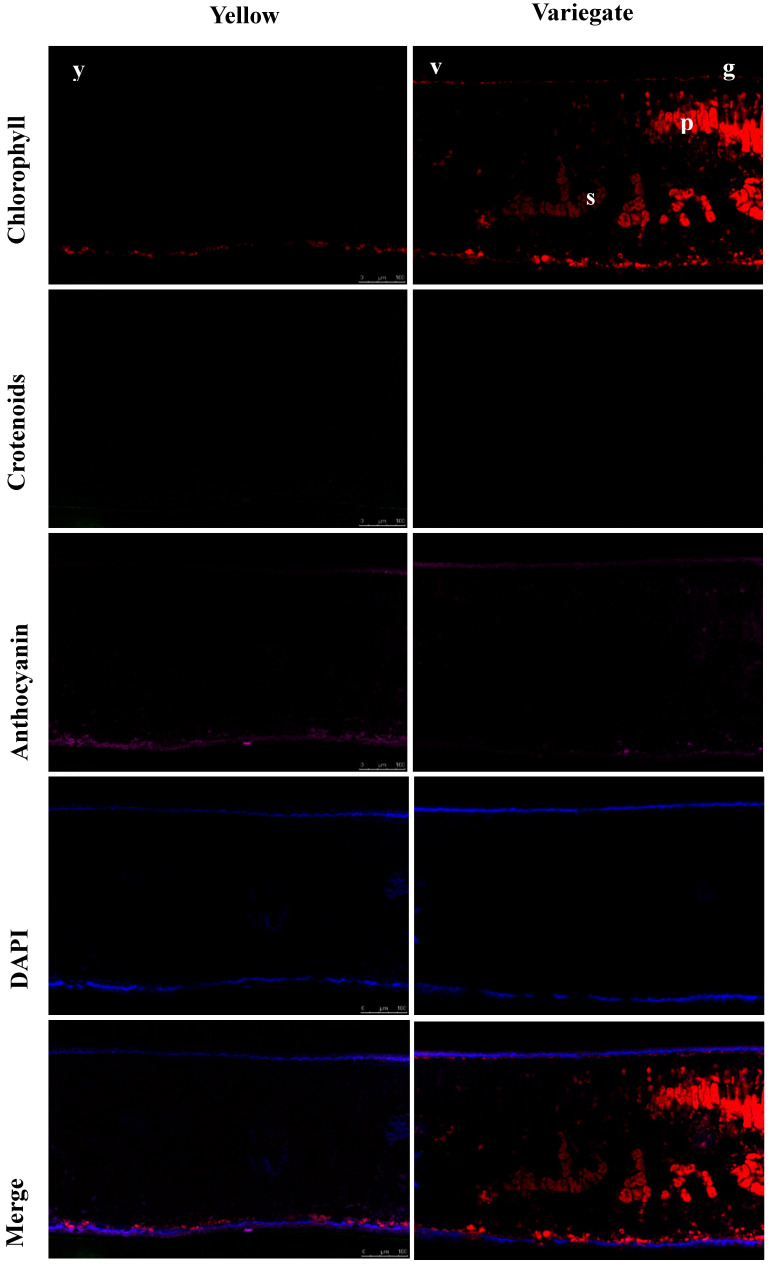
Tissue localization of chlorophyll, carotenoids, and anthocyanins in the leaves analyzed by confocal laser-scanning microscopy. p: palisade parenchyma; s: spongy parenchyma; DAPI: 4′,6-diamidino-2-phenylindole; DIC: differential interference contrast; y: full-yellow juvenile leaf; v: yellow sectors of variegated leaf; g: green sectors of variegated leaf. Scale bar = 100 μm.

**Figure 4 plants-10-00552-f004:**
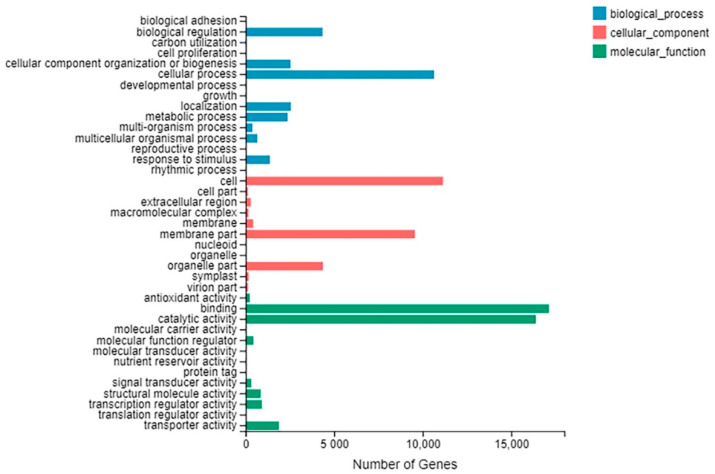
GO classification of unigenes. All the unigenes annotated were classified into three main categories: biological process, cellular component and molecular function.

**Figure 5 plants-10-00552-f005:**
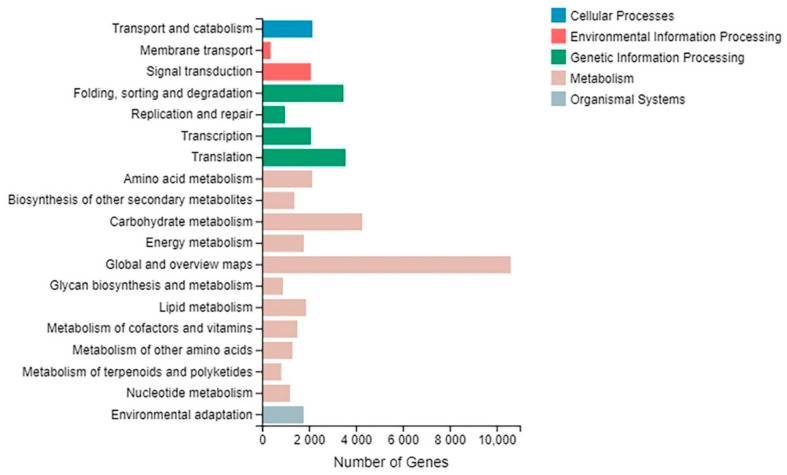
The KEGG pathway classification of assembled unigenes. The *x*-axis indicates the number of unigenes and *y*-axis indicates the function classes.

**Figure 6 plants-10-00552-f006:**
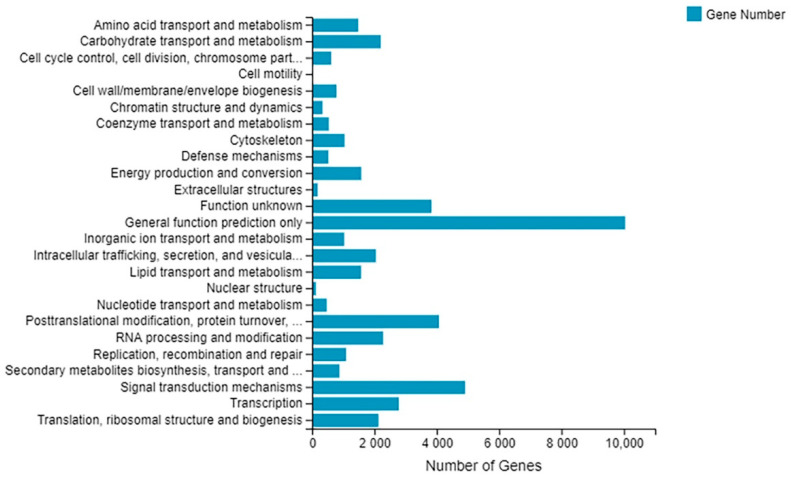
Histogram of KOG function classification of unigenes. The categories of the KOG are shown on *y*-axis and the *x*-axis represents the number of unigenes.

**Figure 7 plants-10-00552-f007:**
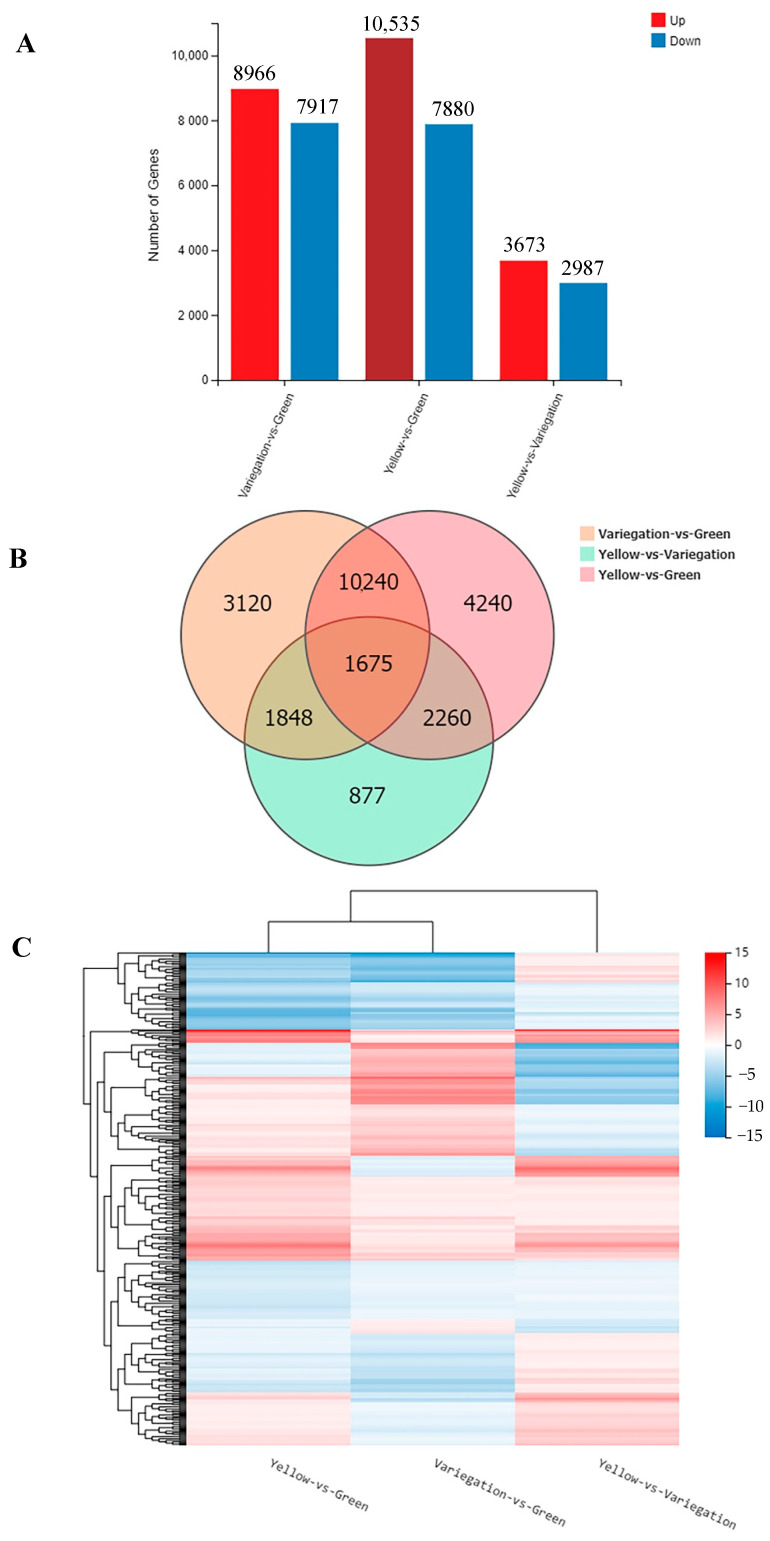
Statistical and clustering analysis of differentially expressed genes. (**A**) Statistic of differentially expressed genes. Upregulated genes were shown in red and downregulated genes were shown in blue. (**B**) Venn diagram of differentially expressed genes. Numbers in the overlapping regions indicate genes differentially expressed in more than one pairwise comparison. (**C**) Clustering heat map of differentially expressed genes. Red color in the figure represents the highly expressed genes and blue color represents the genes expressed at low levels.

**Figure 8 plants-10-00552-f008:**
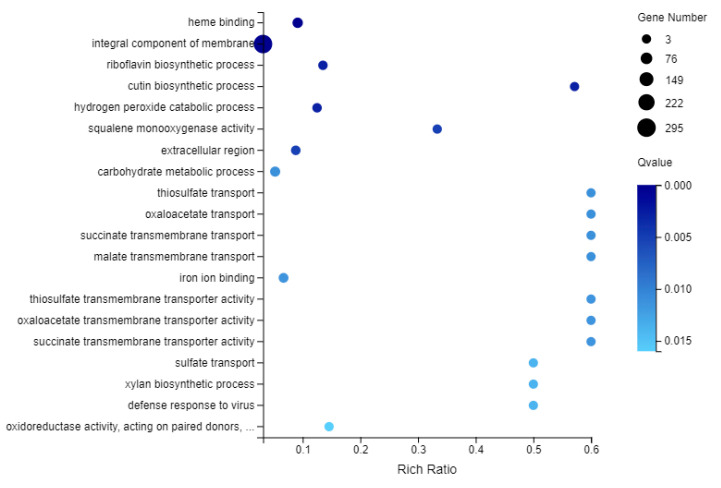
GO enrichment of DEGs.

**Figure 9 plants-10-00552-f009:**
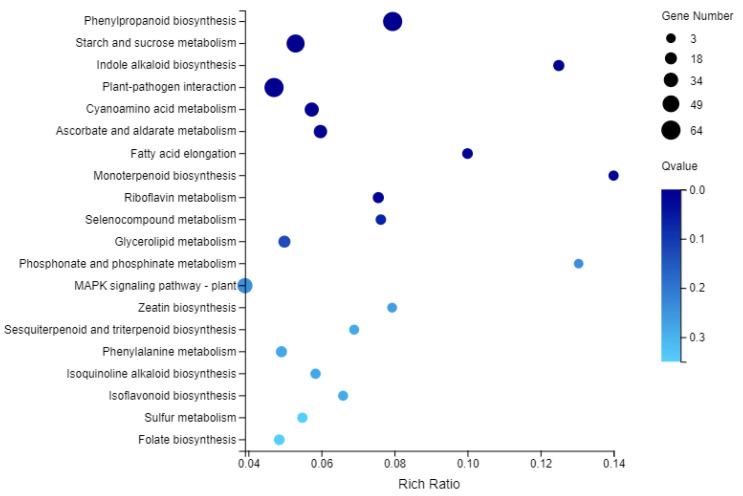
A bubble plot of KEGG pathway enrichment of DEGs showing the top 20 enriched pathways. The size of dots represents the number of genes. MAPK, mitogen-activated protein kinase.

**Figure 10 plants-10-00552-f010:**
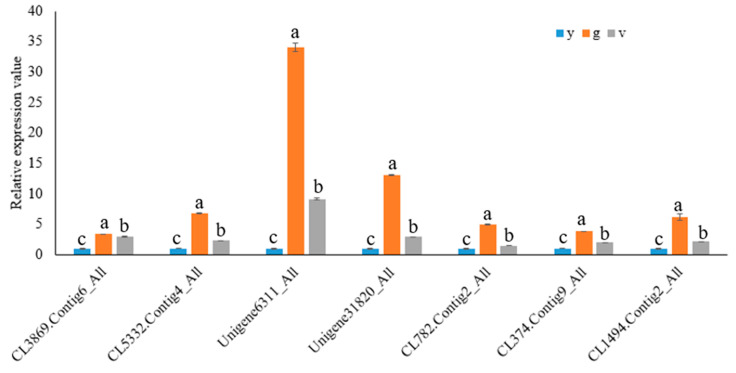
Quantitative real-time PCR verification of RNA-Seq data. Seven DEGs were selected for the verification of the reliability of the transcriptome sequencing results. The expression of β-actin was used as an internal reference. Data represent means ± SEM of three independent experiments. Different lowercase letters (a, b and c) indicated a significant difference among yellow leaves (y), green sectors of the variegated leaves (g) and yellow sectors of the variegated leaves (v) at *p*  <  0.05 level. a: significant difference compared to yellow sectors of variegated leaves; b: significant difference compared to green sectors of variegated leaves; c: significant difference compared to juvenile yellow leaves.

**Table 1 plants-10-00552-t001:** Pigment contents in yellow leaves and green and yellow sectors of variegated leaves.

	Yellow	Green	Variegation
Chl a (mg·g^−1^ FW)	0.09 ± 0.01 ^b^	1.28 ± 0.09 ^a^	0.11 ± 0.01 ^b^
Chl b (mg·g^−1^ FW)	0.05 ± 0.01 ^c^	0.38 ± 0.01 ^a^	0.10 ± 0.01 ^b^
Chl a+b (mg·g^−1^ FW)	0.14 ± 0.01 ^b^	1.66 ± 0.10 ^a^	0.21 ± 0.01 ^b^
Chl a/b	1.63 ± 0.07 ^b^	3.33 ± 0.12 ^a^	1.14 ± 0.03 ^c^
Car (mg·g^−1^ FW)	0.05 ± 0.01 ^b^	0.46 ± 0.03 ^a^	0.06 ± 0.01 ^b^
Chl a+b/Car	2.91 ± 0.12 ^c^	3.65 ± 0.04 ^a^	3.31 ± 0.09 ^b^
Ant (mg·g^−1^ FW)	0.10 ± 0.01 ^b^	0.145 ± 0.01 ^a^	0.10 ± 0.01 ^b^

Data presented were means ± SEM from three replicates. Different superscript lowercase letters (a, b and c) in each row indicated a significant difference among yellow leaves (yellow), green sectors of the variegated leaves (green) and yellow sectors of the variegated leaves (variegation) at *p*  <  0.05 level. a: significant difference compared to yellow sectors of variegated leaves; b: significant difference compared to green sectors of variegated leaves; c: significant difference compared to juvenile yellow leaves. Chl a: chlorophyll a; Chl b: chlorophyll b; Chl a+b: total chlorophyll (chlorophyll a+b); Chl a/b: the ratio of chlorophyll a/chlorophyll b; Car: carotenoid; Chl a+b/Car: the ratio of total chlorophyll/carotenoid; Ant: anthocyanin; FW: fresh weight.

**Table 2 plants-10-00552-t002:** Functional annotation of the unigenes in different databases.

Database	Number of Annotated Unigenes	Percentage of Annotated Unigenes
NR	59,752	75.98%
NT	52,105	66.26%
Swiss-Prot	45,066	57.31%
KEGG	47,967	61.00%
KOG	47,369	60.24%
Pfam	46,101	58.63%
GO	34,793	44.25%
Intersection	20,462	26.02%
Overall	62,275	79.19%

**Table 3 plants-10-00552-t003:** Putative structure genes involved in chlorophyll metabolism and chloroplast integrity and function.

Gene ID	Log2 Fold Change			Annotation
	Variegation vs. Green	Yellow vs. Green	Yellow vs. Variegation	
CL5332.Contig4_All	3.20	2.16	−1.04	magnesium-protoporphyrin IX monomethyl ester [oxidative] cyclase
CL374.Contig9_All	8.64	3.97	−4.66	chlorophyll(ide) b reductase
CL8272.Contig2_All	7.34	2.73	−4.61	chlorophyll(ide) b reductase
CL8272.Contig3_All	−3.41	−2.15	1.26	chlorophyll(ide) b reductase
Unigene31820_All	8.64	2.79	−5.85	monogalactosyldiacylglycerol synthase
CL3869.Contig6_All	2.09	8.83	6.73	digalactosyldiacylglycerol synthase

## Data Availability

The data presented in this study are available on request from the corresponding author.

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
