# Peer review of "Cytological and Transcriptomic Analysis Provide Insights into the Formation of Variegated Leaves in Ilex × altaclerensis ‘Belgica Aurea’"

_plants, 2021, doi:10.3390/plants10030552_

Round 1
Reviewer 1 Report
In nature, foliar variegation is a defensive coloration, which, in particular, protects leaves from the herbivore insects. The problem of the mechanisms underlying the variegation of plants is actual one. In order to elucidate the regulatory mechanisms for leaf variegation, leaf structure, pigment content and transcriptome analysis were compared between juvenile yellow leaves and variegated leaves of I. altaclerensis ‘Belgica Aurea’ in the present study. This manuscript is well written and contains reasoned conclusions. It could be published after small correction. I have several specific comments.
P. 2 lines 83-84. It is necessary to refer to Fig. 1A and C. Information about the different types of leaves there are in paragraph 4.1. only (P. 14 line 388).
Fig. 2. Chloroplast images in transmission microscope are low-contrast. This is oddly enough because usually chloroplasts are fixed satisfactorily. I suggest replacing the photos with the best ones.
In paragraph 4.3., details of a method (temperature, duration of all stages, pH of the solutions, etc.) are absent.
Author Response
Dear Editor
We would like to thank the referees for their valuable comments and suggestions. We have revised our manuscript carefully according to the referees’ suggestions. Please see the attached.

Reviewer 2 Report
Authors of the manuscript no plants-1102222 have performed the cytological and transcriptome analysis of the juvenile leaves and mature variegated leaves of Ilex × altaclerensis ‘Belgica Aurea’ (in green and yellow sectors of leaves, separately). Using TEM analysis they were able to observe differences in chloroplast ultrastructure between juvenile yellow leaves, yellow sectors and green sectors of the variegated leaves. Chlorophyll content analysis showed that the leaf variegation formed in I. altaclerensis ‘Belgica Aurea’ was a total chlorophyll deficient type. Using the RNA-seq technology authors were able to identify 1675 DEGs in more than one pairwise comparison, among which genes encoding for enzymes from chlorophyll biosynthesis as well as MGDG and DGDG synthases were found.
The manuscript meets the criteria for publication in the Journal. Regarding the experimental part of work, the authors have used advanced and suitable methodology, including bioinformatics tools and analysis. The manuscript is written with adequate English; minor linguistic issues are listed below.
However, there are several major and minor issues which should be addressed or explained before considering this work for publishing in Plants.
Major points:
- Section 4.9: Regarding the qRT-PCR analysis – the method used for calculation of relative expression levels was not up to standards. Why the Livak method was used instead of the Pfaffl method? Did authors validated the efficiency of qRT-PCR reaction for each pair of primers and compared them? For efficiencies varying by less than 10% the Livak method can be used, if the experimentally established efficiencies vary by more than 10%, a correction should be done using a Pfaffl mathematical model (Pfaffl, M.W., 2001. A new mathematical model for relative quantification in real-time RTPCR. Nucleic Acids Res. 29, 16–21.). Please also refer to: Eleven golden rules of quantitative RT-PCR (Udvardi et al., 2008). The MIQE guidelines for real-time PCR experiments should be followed (Bustin et al., 2009). Also, the melting curve (sometimes called dissociation curve) analysis should be used for assessment of the specificity of real-time PCR reactions performed with the SYBR Green dye; was it done for each amplification product?
- Section 4.6: Explain how the plant harvesting for RNA isolation was performed? What does the “biological replicate” mean? How many plant individuals were harvested per one biological replicate? Provide more details regarding the magnetic beads, first and second strand synthesis, the addition of adapters and final PCR amplification – add names of the products and names of manufacturers.
- Table 1: the footnote lacks the explanation of letters a, b and c, used in the table for statistics. Make the legend more informative. Also, explain abbreviations used in the table such as: Chl, Car, Ant.
- Section 2.2: Is not clear for the reader, the text does not correspond clearly with the Table 1, probably also due to missing explanations of the statistics (mentioned above).
- Section 2.9 and Table 3: In the text (also in section 2.10 and Discussion) authors mention 7 DEGs, while Table 3 lists 6 DEGs. Please, correct it. Also, please, explain why two unigenes (CL374.Contig9_All and CL8272.Contig2_All) encoding chlorophyll(ide) b reductase were down-regulated in yellow leaves and yellow parts of the variegated leaves, while one unigene (CL8272.Contig3_All) encoding this enzyme was up-regulated. Include your explanation in the Discussion section.
- Section 2.10: Out of 7 enzymes which expression was analysed in this section (present in Figure 10), four are present as DEGs in the Table 3 and three are not. Why is that? Are these the same 7 DEGs as those mentioned in the section 2.9? This is very misleading and unclear for the reader! Please select the DEGs and stick to your selection.
- Additionally, you didn’t perform any statistical analysis of the data from the Figure 10. Hence, we do not know, which changes are significant and which are not. Given the huge error bars of the relative expression values in the green tissue, differences in expression between green and yellow tissues will probably not be indicated as significant in any of the statistical analysis, in most of the unigenes analysed. Provide statistical analysis and make new conclusions accordingly. In the current version, your statement: “The qRT-PCR data showed that the relative expression levels of all 7 DEGs increased in the green parts of the variegated leaves” is not justified by the results.
- You lists 7 DEGs encoding for enzymes clearly involved in either chlorophyll or thylakoid membrane lipid biosynthesis. However, you have produced a substantial amount of data through RNA-seq and would be unfortunate to overlook important findings. What are the most up and down regulated genes? Please analyse the very top and bottom of your list. Are there any other (less obvious) conclusions you could make?
Minor points:
- Line 35: delete “and so on”
- Line 58: recent
- Line 60: japonicus, In plant species other than Arabidopsis…. (delete first “other”)
- Line 68: belonging
- Line 69: …which bears….
- In autumn and winter I. altaclerensis ‘Belgica Aurea’ has sparse red berries on the branches.
- Figure 2: make A, B, C font bigger
- Figure 3: P and S capital or small? Cross-check symbols on the figure with the legend. Explain: g, y, v.
- Lines 311-312: provide references
- Line 320: However instead of but
- Line 457: samples instead of treatments
Author Response
Dear Editor,
We would also like to thank the referees for their valuable comments and suggestions. We have revised our manuscript carefully according to the referees’ suggestions. Please see the attached.

Round 2
Reviewer 2 Report
Dear Authors,
Compared to its previous version, I find the current manuscript improved. You have corrected all minor and answered most major issues. Below, I refer to major issues I find not fully answered by your recent revision:
- Even using the top-quality equipment does not substitute for calculating the reaction efficiency. By using default settings of your real-time PCR system, you have set the efficiency of the reaction by 2.00. By using the formula: 2-ΔΔCt you ASSUMED that each pair of your primers performs in reactions with the efficiency = 100%, meaning that after each PCR cycle the amount of the amplified product EXACTLY DOUBLES. This is rarely true. In the above formula, 2 stands for 1+E, where E is the REAL reaction efficiency, which largely depends on the primer performance. Therefore, you should have empirically checked the efficiency of each of your primer pair by performing reactions with standard cDNA, in a 4 to 6-point concentration series. The slope of each curve indicates the efficiency of respective primer pair, which you should have substituted with E in the above formula, to provide the efficiency correction (according to Pfaffl method). You do not need to establish the reaction efficiency if you: a) use commercially available primers (this is for example the case for Taq Man assays) b) if you are using primers already published with known efficiency. If you have designed the primers yourself and are using them for the first time – you should have performed the above experiment to know your REAL reaction efficiency (E). However, I can often see in various publications the use of the Livak method, which to my opinion is very unfortunate and not up to standards, which I have mentioned previously. Therefore, regarding the calculation method in your quantitative real-time PCR analysis, I will leave the final decision to the Editor.
3 and 7. Regarding Table 1 and Figure 10 footnote: I can see you have added the sentence explaining letters a, b and c, used for statistics. However, to my opinion it is still not clear for the user, what is the meaning of each letter. If this is not the p-value (as you have chosen one p-value level: p<0.05) then it must be the difference compared to a certain tissue/sample. I would suggest putting it this way (just guessing):
a – significant difference compared to yellow sectors of variegated leaves
b - significant difference compared to green sectors of variegated leaves
c - significant difference compared to juvenile yellow leaves
- You have left this question not answered. It is a pity, as you have produced a substantial amount of data through RNA-seq. It would be interesting to know, which genes are the top up- and down-regulated in your comparisons, even if they are not known to be directly related to either chlorophyll or thylakoid membrane lipid biosynthesis. Maybe you would find something novel? I leave it to your decision, whether or not you would like to share these findings to the reader.
Author Response
Reviewer #2
- Dear reviewer, thank you very much for your good suggestion about the quantitative real-time PCR analysis. To test the efficiency of respective primer pair we used in our paper, we have to perform a second round of quantitative real-time PCR analysis. So, we have to collect leaf samples to isolate RNA and then conduct qRT-PCR. The problem is that the new leaves do not sprout, we have to stay for several months to collect the leaves.
- Thanks for your good suggestion. To make it clear for the readers, we have added the explanation of a, b and c in the revised paper. And the following sentences that you suggested have been added in the footnotes of Table1 and Figure 10.
“a, significant difference compared to yellow sectors of variegated leaves; b, significant difference compared to green sectors of variegated leaves; c, significant difference compared to juvenile yellow leaves.”
- We are sorry that we have misunderstand your comment. We thought that your comment “What are the most up and down regulated genes?” only refer to the DEGs listed in Table 3.
We analyzed the 1675 DEGs, and found that the most upregulated gene in green sectors of the variegated leaves is Unigene30740_All, which encodes the DNA-directed RNA polymerase alpha subunit (chloroplast). This may suggest that the elevated expression of RNA polymerase mRNA in chloroplasts is involved in chloroplast biogenesis. The most downregulated gene in green sectors of the variegated leaves is CL222.Contig6_All encoding NAD(P)H-quinone oxidoreductase subunit 2. Why this gene is downregulated needs further investigation.
The following sentences have been added in the revised paper. “Furthermore, analysis of the 1675 DEGs revealed that the most upregulated gene in green sectors of the variegated leaves was Unigene30740_All, which encoded the DNA-directed RNA polymerase alpha subunit (chloroplast) (Table S5). This may suggested that the elevated expression of RNA polymerase mRNA in chloroplasts was involved in chloroplast biogenesis. The most downregulated gene in green sectors of the variegated leaves was CL222.Contig6_All encoding NAD(P)H-quinone oxidoreductase subunit 2. The reason of the downregulation of this gene needs further investigation.”
Thank you again for your efforts and time.
Sincerely yours,
Prof. Min Zhang